# Picomolar SARS-CoV-2 Neutralization Using Multi-Arm PEG Nanobody Constructs

**DOI:** 10.3390/biom10121661

**Published:** 2020-12-11

**Authors:** Ainhoa Moliner-Morro, Daniel J. Sheward, Vivien Karl, Laura Perez Vidakovics, Ben Murrell, Gerald M. McInerney, Leo Hanke

**Affiliations:** 1Department of Microbiology, Tumor and Cell Biology, Karolinska Institutet, 17177 Stockholm, Sweden; ainhoa.moliner.morro@ki.se (A.M.-M.); daniel.sheward@ki.se (D.J.S.); vivien.karl@ki.se (V.K.); laura.perez.vidakovics@ki.se (L.P.V.); benjamin.murrell@ki.se (B.M.); gerald.mcinerney@ki.se (G.M.M.); 2Division of Virology, Institute of Infectious Diseases and Molecular Medicine, Faculty of Health Sciences, University of Cape Town, 7925 Cape Town, South Africa

**Keywords:** single-domain antibody fragment, nanobody, neutralization, sortase A, click chemistry, PEG linker, multivalent, SARS-CoV-2

## Abstract

Multivalent antibody constructs have a broad range of clinical and biotechnological applications. Nanobodies are especially useful as components for multivalent constructs as they allow increased valency while maintaining a small molecule size. We here describe a novel, rapid method for the generation of bi- and multivalent nanobody constructs with oriented assembly by Cu-free strain promoted azide-alkyne click chemistry (SPAAC). We used sortase A for ligation of click chemistry functional groups site-specifically to the C-terminus of nanobodies before creating C-to-C-terminal nanobody fusions and 4-arm polyethylene glycol (PEG) tetrameric nanobody constructs. We demonstrated the viability of this approach by generating constructs with the SARS-CoV-2 neutralizing nanobody Ty1. We compared the ability of the different constructs to neutralize SARS-CoV-2 pseudotyped virus and infectious virus in neutralization assays. The generated dimers neutralized the virus similarly to a nanobody-Fc fusion variant, while a 4-arm PEG based tetrameric Ty1 construct dramatically enhanced neutralization of SARS-CoV-2, with an IC_50_ in the low picomolar range.

## 1. Introduction

In addition to conventional antibodies, camelids produce heavy chain-only antibodies, which lack the light chain [1]. The antigen-binding region of these antibodies is composed of a single domain which can be expressed independently as a ~15 kDa antibody fragment and is referred to as a VHH or nanobody [2,3]. Despite being only 10% of the size of a conventional IgG antibody, and despite their monomeric nature, nanobodies can bind their target antigen with high specificity and affinity. Nanobodies are biochemically well-behaved proteins and can be expressed to high levels in yeast, *E. coli*, or mammalian cells [4,5]. Because of their small size, they are relatively easy to functionalize using genetic, chemical, or enzymatic strategies. Unsurprisingly, nanobodies are increasingly popular in molecular biology research, molecular imaging, immunology, as diagnostic tools or for therapeutic applications [6,7,8,9,10,11].

As virus neutralizing agents, nanobodies are interesting candidates because they may be able to target viral epitopes that are inaccessible to conventional antibodies, providing unique opportunities for antiviral intervention. Nevertheless, the monomeric nature of nanobodies can also be of disadvantage. Binding kinetics, especially dissociation rates (k_off_) can limit neutralization potency. Potency can be substantially increased through dimerization, which enhances avidity and reduces dissociation rates. This is often achieved by nanobody-IgG Fc fusion, but other strategies to expand valency may also be suitable. An example is the fusion to the B-subunit of *E. coli* verotoxin for nanobody pentamerization [12]. Other examples include genetic fusion with individual binding domains separated by glycine-serine linkers of different length [13,14,15,16,17]. Such genetically linked nanobody constructs can also be fused to IgG1-Fc domains as recently described influenza neutralizing multidomain constructs illustrate [18].

Sortase A from *Staphylococcus aureus* is a transpeptidase that mediates the ligation of an oligoglycine nucleophile to a LPXTG-containing substrate [19]. Because of its ease of use, mild reaction conditions, and site-specificity, sortase labeling is an attractive method for protein functionalization [20]. The suitability of this labeling approach for functionalization of nanobodies has been demonstrated over recent years. Examples are the conjugation of peptides [11], biotin and fluorophores [21,22], or probes for positron emission tomography [23,24], among others. However, the requirement for custom synthesis of the oligoglycine nucleophiles conjugated to functionalizing elements can be limiting. In addition to oligoglycine nucleophiles, improved sortase A enzymes and reaction conditions also allow the attachment of non-amino acid alkylamines [25,26]. Although such reactions are typically less efficient, the lower efficiency can be compensated for by using optimized reaction conditions and a large excess of the nucleophile component.

Earlier, we reported the development of an alpaca-derived nanobody, Ty1, which binds to the SARS-CoV-2 spike receptor binding domain (RBD), potently neutralizing the virus [27]. The Ty1 binding site overlaps with the host cell receptor ACE2 binding site and potently neutralizes the virus by directly preventing host receptor attachment. Ty1 can bind the RBD in both the ACE2 accessible “up” and the ACE2 inaccessible “down” conformations and three nanobody monomers can bind to one trimeric spike protein simultaneously, providing opportunities for multivalent neutralization.

Here, we generated a series of multivalent constructs of this nanobody using a combination of sortase A functionalization and click chemistry and evaluated their neutralization potency. We first attached click-chemistry functional groups site-specifically to the C-terminus of the nanobody Ty1 before using strain-promoted azide-alkyne cycloaddition (SPAAC) [28] to generate C-to-C terminal Ty1 dimers and to attach Ty1 to multi-arm polyethylene glycol (PEG) scaffolds. We evaluated the different constructs in SARS-CoV-2 pseudovirus neutralization assays as well as infectious SARS-CoV-2 plaque reduction neutralization tests (PRNTs). We showed that bivalent constructs are significantly more potent than monovalent Ty1, and a 4-arm PEG tetramer equipped with four Ty1 molecules is extremely potent, capable of neutralizing SARS-CoV-2 with an IC_50_ in the low picomolar range.

## 2. Materials and Methods

### 2.1. Protein Production and Purification

#### 2.1.1. Ty1

The sequence encoding the RBD-specific and SARS-CoV-2 neutralizing nanobody Ty1 [27] was cloned into the pHEN plasmid with a C-terminal sortase recognition sequence (LPETG) followed by a 6 × HIS tag (C-terminus: Ty1-GG**LPETG**GHHHHHH). This plasmid was used to transform *E. coli* (BL21) cells for Ty1 expression in the bacterial periplasm. Cells were grown in terrific broth (TB) media and expression was induced with 1 mM isopropyl-β-D-thiogalactopyranoside (IPTG) at OD600 = 0.8 followed by overnight growth at 30 °C. The bacterial pellet from a one-liter culture was resuspended in 10 mL TES buffer (200 mM Tris pH 8, 0.65 mM EDTA, 0.5 M sucrose) for 2 h before adding 25 mL of 0.25× TES buffer and incubating under agitation overnight. The periplasmic extract was separated from cells by centrifugation and added to 2–4 mL of Ni-NTA resin in a gravity flow column. The resin was washed with 20 mL of 50 mM Tris, 150 mM NaCl, and 20 mM imidazole. The protein was eluted in the 50 mM Tris pH 7.5, 150 mM NaCl, and 250 mM imidazole and purified by size-exclusion chromatography on a sephacryl S-200 HR column (Cytiva, Uppsala, Sweden) in 50 mM Tris pH 7.5, 150 mM NaCl.

#### 2.1.2. Sortase A 5M

The plasmid pet30b-5M SrtA was a gift from Hidde Ploegh, Boston Children’s Hospital, Boston, USA, (Addgene plasmid # 51140) [29] and was used to transform BL21 cells for cytoplasmic expression. Expression was induced with 1 mM IPTG at OD600 = 0.8; cells were grown in lysogeny broth (LB) media for 4 h at 30 °C. The protein was extracted by incubation with 150 mM NaCl, 50 mM Tris, protease inhibitor, lysozyme, and DNase, followed by sonication. Lysate was clarified by centrifugation and sortase A was purified by Ni-NTA affinity and size-exclusion chromatography as described for the nanobody.

#### 2.1.3. Ty1-Fc

For mammalian expression of Ty1-Fc, the sequence encoding the nanobody Ty1 was cloned upstream of a human IgG1-Fc. The plasmid was used to transiently transfect FreeStyle 293F cells using the FreeStyle MAX reagent (Life Technologies, Carlsbad, CA, USA) in accordance with the manufacturer’s instructions. The Ty1-Fc fusion was purified from filtered supernatant on Protein G Sepharose (Cytiva, Uppsala, Sweden), eluted in 0.2 M glycine pH 2.2, and purified by size-exclusion chromatography using a Superdex S200 16/600 column (Cytiva, Uppsala, Sweden) in 50 mM Tris pH 7.5 and 150 mM NaCl.

### 2.2. Sortase-Mediated Click-Chemistry Functionalization of Ty1

Ty1 was functionalized site-specifically on the C-terminus using sortase A with either an azide or a dibenzocyclooctyne (DBCO). To functionalize Ty1 with DBCO, 70 µM of Ty1 was incubated with 5 µM sortase A, 8 mM DBCO-amine (Sigma–Aldrich, Saint Louis, MO, USA, #761540), in 150 mM NaCl, 50 mM Tris pH 7.5, and 10 mM CaCl_2_ for 3 h at 25 °C. To functionalize Ty1 with an azide, 70 µM of Ty1 was incubated with 5 µM sortase A, 10 mM 3-Azido-1-propanamine (Sigma–Aldrich, Saint Louis, MO, USA, #762016), in 150 mM NaCl, 50 mM Tris pH 7.5, and 10 mM CaCl_2_ for 3 h at 25 °C. In both reactions, unreacted nanobody, sortase A, and excess nucleophile were removed using Ni-NTA resin and PD-10 desalting columns.

All click reactions were performed in 50 mM Tris pH 7.5 and 150 mM NaCl at 4 °C with functionalized nanobody concentrations ranging from 30 µM (for the dimers) to 90 µM (4-arm PEG). The Ty1 dimer was generated by a Cu-free strain-promoted azide-alkyne click chemistry (SPAAC) reaction by incubating DBCO and azide functionalized Ty1 in a 1:1.2 molar ratio for 24 h.

The Ty1 PEG dimer was generated by SPAAC using Ty1-azide and bis-dPEG11-DBCO (Sigma–Aldrich, Saint Louis, MO, USA, #QBD11372) by adding bis-dPEG11-DBCO in four equal steps (at 0, 0.5, 1.5, and 24 h) to a final 2:1 molar ratio. The reaction was incubated for 48 h.

The tetramer was generated with a SPAAC reaction using an 8:1 molar ratio of DBCO functionalized Ty1 and 4-arm PEG10K-azide (Sigma–Aldrich, Saint Louis, MO, USA, #JKA7163) with a pentaerythritol core structure and an average total molecular weight of 10 k. The molecular weight for the 4-arm PEG corresponded to the sum of the PEG molecular weights of each arm and the number of ethylene oxide units in the PEG chain may not have been equal for all arms. The 4-arm PEG Ty1 reaction was incubated for four days. All constructs generated by click chemistry were purified from unreacted monomeric nanobody by size-exclusion chromatography on a Superdex S200 16/600 (Appendix A). Nanobody constructs were analyzed for size, integrity, and purity by SDS-PAGE (4–12% NuPAGE Bis-Tris, (Life Technologies, Carlsbad, CA, USA) using a PageRuler plus protein ladder (Life Technologies, Carlsbad, CA, USA) and Coomassie G-250 staining.

### 2.3. Pseudovirus Neutralization Assay

Pseudotyped viruses were generated as described earlier [27]. Pseudotyped viruses sufficient to generate ~100,000 relative light units (RLUs) were incubated with serial dilutions of nanobody constructs for 60 min at 37  °C. Approximately 15,000 HEK293T-ACE2 cells were then added to each well and the plates were incubated at 37  °C for 48 h. Luminescence was then measured using Bright-Glo (Promega, Madison, WI, USA) per the manufacturer’s instructions on a GM-2000 luminometer (Promega, Madison, WI, USA) with an integration time of 0.3 s.

### 2.4. Plaque Reduction Neutralization Test

VeroE6 cells (ATCC-CRL-1586) were seeded at 1 × 10^5^ cells/well in a 24-well plate and incubated overnight at 37 °C, 5% CO_2_. A 3-fold serial dilution of the nanobody constructs were made in 96-well plates and were incubated with 100 plaque-forming units (PFU) of a clinical isolate of SARS-CoV-2 [30] for 1 h at 37 °C. Control wells without nanobody constructs were included as well. A nanobody–virus mixture was added to the cells and incubated at 37 °C for 1 h. Cells were washed with PBS and an overlay of 1% CMC/DMEM supplemented with 2% fetal bovine serum (FBS) and antibiotic antimycotic solution (#A5955, Sigma–Aldrich, Saint Louis, MO, USA) was added onto the cells. Three days post infection, cells were fixed in 10% (*v*/*v*) formaldehyde overnight, followed by staining with crystal violet. Plaques were counted, and inhibition was calculated with the following equation:(1)$$\mathrm{Inhibition}\;\left(\%\right)=\left(1-\left(\frac{\mathrm{treated}\;\mathrm{cells}\;\mathrm{pfu}/\mathrm{mL}}{\mathrm{virus}-\mathrm{control}\;\mathrm{pfu}/\mathrm{mL}}\right)\right)\times 100$$

## 3. Results

### 3.1. Sortase-Catalyzed Functionalization of the Nanobody Ty1

Sortase A catalyzes the ligation of oligoglycine containing peptides to the LPXTG motif at or near the C-terminus of proteins, including nanobodies [19]. As substrate surrogates, alkylamines can sometimes be used instead of oligoglycines [25,31]. Here we used the sortase A enzymatic approach to site-specifically install azides and cyclooctynes to the C-terminus of a SARS-CoV-2 neutralizing nanobody Ty1 [27]. The functionalized nanobodies then served as building blocks for the generation of multivalent constructs.

First, we functionalized the nanobody Ty1 (Figure 1A) by attaching a dibenzocyclooctyne-amine (DBCO-NH2) to the C-terminus of Ty1. This reaction was slower than the attachment of oligoglycine containing nucleophiles, but could be performed efficiently using high amounts of the DBCO-amine nucleophile (8–10 mM). Similarly, Ty1 was functionalized with an azide using 3-azido-1-propanamine as nucleophile. After reactions of 2–3 h at 25 °C, sortase A and unreacted nanobody was removed using Ni-NTA agarose, and the labeled nanobody was purified from excess nucleophile on PD-10 desalting columns. As this process removed also all unreacted nanobody (still HIS-tagged), we expected that the purified product only contained the DBCO- or azide-labeled nanobody. The azide and DBCO nanobody then served as building blocks for the generation of various multivalent constructs. Mixing of DBCO and azide containing nanobody in a suitable buffer was then sufficient to initiate strain-promoted azide-alkyne cycloaddition.

### 3.2. Generation of Multivalent Nanobody Constructs

For the dimer (Figure 1B), we incubated Ty1-DBCO and Ty1-azide in a 1:1.2 molar ratio. This reaction allowed for the generation of unnatural C-to-C-terminal nanobody fusions. After 24 h, approximately 30–60% of the protein had dimerized, and the dimer was separated from the unreacted, monomeric Ty1 by size-exclusion chromatography (Appendix A).

To evaluate the effect of linker length on neutralization activity of the dimer, we generated a second Ty1-dimer using a PEG11 linker (Figure 1C). This ~3 nm linker slightly increased the distance between dimerized nanobody molecules. This second dimer was generated by combining azide-functionalized Ty1 with bis-PEG11-DBCO in a 2:1 molar ratio. The click reaction was performed at 4 °C for 48 h and the dimer was purified from the monomeric nanobody by size-exclusion chromatography.

To further increase valency, we used click chemistry reactions to attach Ty1 on 4-arm PEG scaffolds (Figure 1D). We generated Ty1-DBCO as described above and incubated it in an 8:1 molar excess ratio with 4-arm PEG10K-azide. While the reaction at 4 °C quickly yielded dimers and trimers, it took time until four nanobodies had attached to each 4-arm PEG molecule. After 100 h reaction, we separated excess Ty1-DBCO by size-exclusion chromatography and combined the first fractions of the multimer elution peak of obtain purified 4-arm-PEG-Ty1. All constructs were analyzed by SDS-PAGE and Coomassie staining (Figure 2B).

### 3.3. Increased Valency Enhances SARS-CoV-2 Neutralization

To determine whether the different Ty1 constructs neutralized SARS-CoV-2, we first employed an in vitro neutralization assay using lentiviral particles pseudotyped with the spike protein of SARS-CoV-2 (Figure 3A). Dimeric Ty1 formulations increased the neutralization IC_50_ more than 150-fold. Ty1-Ty1 and Ty1-PEG-Ty1 dimers performed similarly, with IC_50_s in the range of 125 pM, and were slightly more potent than Ty1-Fc. The increased valency of the 4-arm-PEG Ty1 substantially improved potency to about 13 pM. In contrast, infection by versicular stomatitis virus G (VSV-G) pseudotyped lentivirus was not inhibited by any of the constructs (Appendix A), and addition of the different constructs to cells post infection with SARS-CoV-2 pseudotyped lentivirus had likewise no effect (Appendix A), confirming specific neutralization. While the bivalent constructs were potent with low IC_50_ values, incomplete neutralization was evident even up to concentrations of 100 nM, with neutralization plateauing at approximately 90%. In contrast, the 4-arm construct reached 100% neutralization at concentrations above 200 pM.

To test whether similar neutralization could be observed with infectious, replication competent SARS-CoV-2, we employed plaque reduction neutralization tests (PRNTs, Figure 3B). Monolayers of Vero E6 cells were infected with infectious SARS-CoV-2 preincubated with a dilution series of the multimeric nanobody constructs. Corresponding to the pseudovirus assay, the three dimeric constructs neutralized SARS-CoV-2 at comparable picomolar concentrations. Similarly, the 4-arm-PEG-Ty1 construct showed markedly enhanced neutralization of SARS-CoV-2.

## 4. Discussion

Multivalent antibody constructs have a broad range of clinical and biotechnological applications. Small soluble antibody fragments such as nanobodies are suitable building blocks for these constructs, allowing increased valency while keeping the overall molecule size small. We here describe a novel method for the rapid generation of bi- and multivalent nanobody constructs with oriented assembly by Cu-free click chemistry. We demonstrated that C-to-C-terminal fused nanobody constructs increased SARS-CoV-2 neutralization potency more than 150-fold compared to the monomeric variant. Interestingly, the click-chemistry fused dimers reached 50% virus neutralization at similar concentrations as the nanobody-Fc fusion, suggesting that valency, rather than steric hindrance, was responsible for the increase in potency. The 4-arm PEG-nanobody construct further enhanced the neutralization potency of the bivalent constructs. In addition, neutralization curves rapidly approached 100% for this construct in both assays. Combined, this may indicate additional mechanisms of neutralization. The binding site of Ty1 is accessible on all three protomers of the spike trimer, as revealed by cryo-EM [27]. With an estimated linker length of 30–40 nm between nanobodies on the 4-arm PEG10K construct, it is conceivable that three Ty1 molecules could bind to a single spike trimer concurrently. In this scenario, the fourth Ty1 molecule might enable bridging of spike complexes on virions or even crosslinking of virions to form large immune complexes.

The fusion of full-length antibodies and the generation of chimeric proteins with a combination of sortase catalyzed modification and click chemistry has been shown before [32,33], but they rely on the custom synthesis of oligoglycines conjugated to click-chemistry functional groups. The nucleophiles required for our approach can conveniently be purchased off the shelf. This workflow can in theory also be used to generate other constructs to further increase valency. For example, other multi-arm PEGs (6-arm, 8-arm) are suitable, and other scaffolds including cyclodextrin or PEGylated capsules are also attractive [34,35]. The main advantage of the approach described here over genetic fusion is the oriented assembly, which avoids potentially interfering linkers at the N-terminus near the complementary determining regions (CDRs) of the nanobody. It further allows to start with natively folded proteins, and to rapidly test different nanobody combinations without the requirement for individual cloning and expression.

Nanobodies are ideal molecules for the neutralization of viruses, clearly demonstrated by a large arsenal of antiviral nanobodies reported to date. Nanobodies that target chikungunya virus, influenza virus, HIV, HSV 2, rotavirus, respiratory syncytial virus (RSV), rabies virus, and different coronaviruses, among others, have been reported [27,36,37,38,39]. The main targets for neutralizing antibodies are viral fusogenic glycoproteins which are typically dimeric or trimeric, providing opportunities for multivalent binding. It is thus not surprising, therefore, that multivalent and bispecific constructs can significantly increase neutralizing potency [40,41,42], attributed to higher avidity. A recent example is the nanobody developed by Wrapp et al. [43]. The nanobody VHH-72, originally developed against the RBD of SARS-CoV-1, cross-reacts with the RBD of SARS-CoV-2 but only poorly neutralizes SARS-CoV-2, likely due to a relatively fast dissociation rate (k_off_). Fc-mediated dimerization of VHH-72 and resulting avidity effects largely compensate for the difference in off-rates, allowing VHH-72-Fc to neutralize both viruses.

Increasing avidity can also be achieved by constructing bispecific constructs for simultaneous targeting of different epitopes. Such constructs contain a larger footprint on the viral protein and may thus be less susceptible to viral escape. This is especially relevant when considering that single amino acid escape substitutions often do not completely abolish binding, but impact the dissociation constant [22]. In addition, bi-specific probes could confer a broader neutralization capacity, by neutralizing different strains. The dimeric constructs reported here, based on a click reaction between a DBCO and an azide, would allow the rapid construction and screening of such bispecific nanobody constructs for simultaneous targeting of different epitopes. The PEG-based dimer and the 4-arm PEG nanobody constructs would allow stepwise installation of nanobodies with different specificities when performing click reactions with regulated stoichiometry and intermediate purification steps.

One disadvantage of nanobodies for prevention and treatment of viral disease is their short serum half-life and rapid renal clearance. Though we did not evaluate the serum half-life of 4-arm PEG10K conjugated nanobodies here, PEG conjugation can significantly increase serum half-life of molecules, including antibodies [44,45]. This would provide an additional, significant benefit over and above improved neutralization potency. Using multi-arm PEG of different molecular sizes and valencies may thus allow the careful adjustment of serum half-life and neutralization potency.

## 5. Conclusions

We here described a new rapid method for the generation of bi- and multivalent nanobody constructs using a combination of sortase A functionalization and click chemistry. We generated such constructs with a SARS-CoV-2 neutralizing nanobody and showed that the increased valency substantially increased neutralization potency, leading to IC_50_ values in the low picomolar range.

## Figures and Tables

**Figure 1 biomolecules-10-01661-f001:**
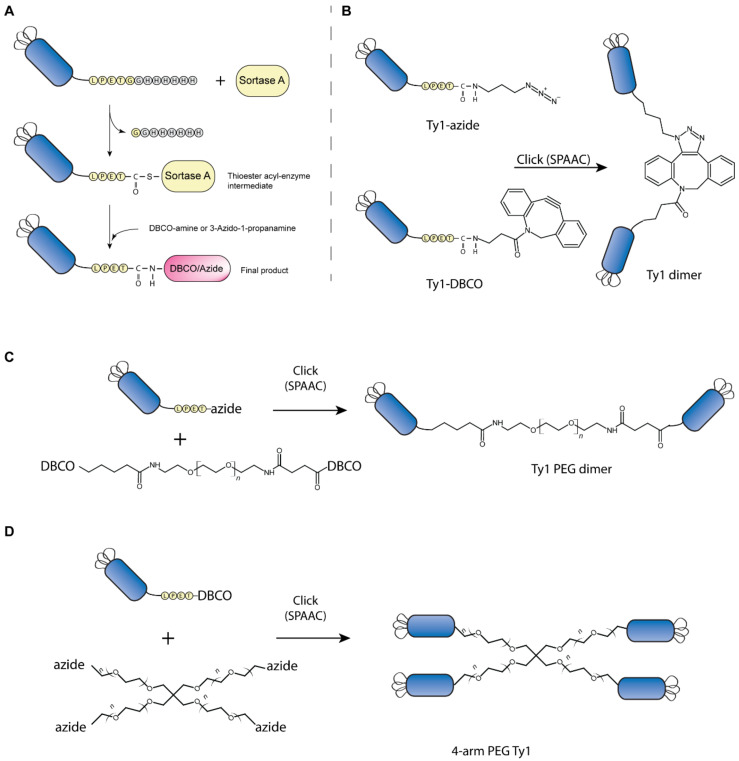
Sortase A mediated functionalization of nanobodies and assembly of multimeric nanobody constructs. (**A**) Sortase A recognizes LPXTG motifs near the C-terminus, cleaves the motif after the threonine and forms a thioester acyl-enzyme intermediate. The C-terminal HIS-tag is cleaved off in the process. The acyl-enzyme intermediate is resolved by the addition of the amine nucleophile DBCO-amine or 3-Azido-1-propanamine, which leads to release of sortase and results in ligation of the molecules to the C-terminus of the nanobody. (**B**) Purified DBCO and azide functionalized nanobodies are incubated for strain-promoted azide-alkyne click chemistry (SPAAC) that results in C-to-C-terminal nanobody fusion and dimer formation. (**C**) Nanobody dimers connected via polyethylene glycol (PEG)-linker molecules are generated by click reactions of azide-labeled nanobodies on bis-PEG-DBCO. (**D**) To expand valency, DBCO-labeled nanobodies are attached to multi-arm PEG-azide molecules. Note that illustrations of constructs and PEG-linkers are not to scale.

**Figure 2 biomolecules-10-01661-f002:**
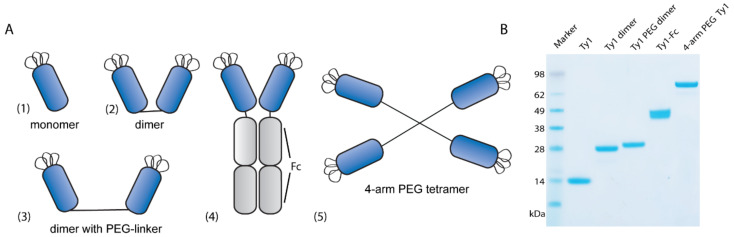
Overview of the generated constructs. (**A**) (1) Ty1 monomer, (2) Ty1-dimer generated as described in Figure 1B, (3) Ty1 dimer with PEG linker, generated by performing a click reaction of Ty1-azide and bis-PEG11-DBCO, (4) Ty1-Fc, dimerization through Fc fusion and expression in mammalian 293F cells [27], (5) 4-arm PEG10K Ty1, generated by performing click reaction of Ty1-DBCO to 4-arm PEG10K-azide. (**B**) Nanobody constructs were loaded on SDS-PAGE and stained using Coomassie G-250. Then 2 ug of each construct was loaded on the gel; molecular weight is indicated on the left (kDa).

**Figure 3 biomolecules-10-01661-f003:**
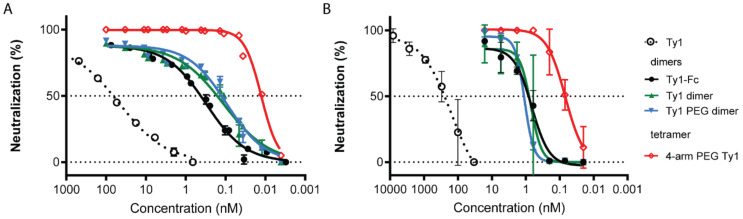
Ty1 multimeric constructs neutralize SARS-CoV-2 spike pseudotyped lentivirus and infectious SARS-CoV-2. (**A**) SARS-CoV-2 spike pseudotyped lentivirus was incubated with a dilution series of Ty1, Ty1-dimers, and 4-arm PEG Ty1, and the percent reduction in infectivity relative to control wells in the absence of inhibitors is shown. (**B**) Dilution series of monomeric Ty1 and the multimeric constructs were incubated with 100 plaque-forming units (PFU) of infectious SARS-CoV-2 for 1 h before infecting monolayers of Vero E6 cells. Plaques were quantified 72 h post infection. Neutralization representing the reduction in the number of plaques relative to control wells is shown. Error bars represent standard deviation (SD) across replicate experiments but are not shown where the size of the error bar is smaller than the symbol size.

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
