# Peer review of "Picomolar SARS-CoV-2 Neutralization Using Multi-Arm PEG Nanobody Constructs"

_biomolecules, 2020, doi:10.3390/biom10121661_

Round 1

Reviewer 1 Report

ARTICLE SUMMARY

In this paper, Moliner-Morro and colleagues present the generation of multivalent constructs of Nanobody Ty1, a camelid single-domain antibody directed against the spike protein of SARS-CoV-2. In order to produce these multivalent constructs, the authors employ the sortase A tag-mediated ligation followed by click chemistry. The Ty1-derived constructs are then tested for their potency to interfere with viral infectivity and the authors successfully demonstrate the improved potency of the multivalent constructs.

DETAILED REPORT – OVERALL CONSIDERATION

I would like to congratulate the authors with the amount of work they have performed and with the care they have taken to compile this manuscript. Overall, the paper is well-written and the work appears to be technically well-executed. The manuscript presents novel, interesting findings that are supported by nice figures. However, some portions of the manuscript remain on the “light” side and more effort should be put into expanding certain sections. I think this will only further improve the quality of the manuscript and provide the interested readers with a very complete description of both the literature and the results. My recommendation would be to accept this manuscript after a round of minor revisions. My comments/suggestions are intended to be as constructive as possible and I hope the authors will read them bearing this in mind.

COMMENTS RELATED TO THE ABSTRACT AND INTRODUCTION

Comment 1.1 (Page 1, Line 29): I believe there should be a comma after “antibodies”.

Comment 1.2 (Page 1, Line 31): The reference molecular mass of Nanobodies is usually ~15 kDa. Please adapt accordingly.

Comment 1.3 (Page 1, Line 35): “E.coli” seems to be lacking a space between the “E.” and “coli” + the following comma and “or” should not be italicized.

Comment 1.4 (Page 1, Line 39): I believe there should be a comma after “agents”.

Comment 1.5 (Page 2, Lines 44 to 45): The authors rightfully state that multi-valency is often achieved through the generation of Nb-IgG Fc fusions, but there are other methods as well and I think the authors should expand on this to make the introduction more complete. Examples include:

Fusing Nbs to the B-subunit of E. coli verotoxin.

Zhang et al 2004 (doi: 10.1016/j.jmb.2003.09.034)

Generating multi-valent Nb-Nb fusions

Pinto Torres et al 2018 (doi: 10.1038/s41598-018-26732-7)

Bernedo-Navarro et al 2018 (doi: 10.3390/toxins10030108)

Desmyter et al 2017 (doi: 10.3389/fimmu.2017.00884)

Generating multi-valent Nb-Nb-Ig Fc fusions

Laursen et al 2018 (doi: 10.1126/science.aaq0620)

The authors should please feel free to include other relevant papers to their liking.

Comment 1.6 (Page 2, Lines 46 to 54): Also, here, I propose to expand the introduction on the use of sortase A. Many sortase A – Nb papers have been published in recent years and I feel that the authors could provide a bit more context.

Examples include

Hagemeyer et al 2015 (doi: 10.1038/nprot.2014.177)

Massa et al 2016 (doi: 10.1002/cmmi.1696)

COMMENTS RELATED TO MATERIALS AND METHODS

With regards to the production and purification procedures, I feel the authors are too restrictive and should provide more experimental details. This will provide interested readers with the full information to reproduce these experiments and appreciate how much work went into this. There is a general trend in research papers where detailed production and purification protocols are too often omitted. Producing and purifying correctly folded and functionally relevant proteins lies at the basis of our craft and requires a lot of hands-on expertise and experience. The inclusion of detailed protocols will better reflect the amount and quality of work performed by the authors.

Comment 2.1 (Page 2, Line 72): given that protein purification is also discussed here I would propose to rename the title to “Protein production and purification”.

Comment 2.2 (Page 2, Lines 73-80): please provide a detailed overview of production and purification methods

Which culture medium was employed for Ty1 production?

How was the periplasmic extraction performed (details)?

How was the IMAC performed (details, buffers, linear gradient vs step gradient)?

Which column was employed for SEC?

Comment 2.3 (Page 2, Line 76): “E.coli” seems to be lacking a space between the “E.” and “coli”

Comment 2.4 (Page 2, Lines 81 to 86): please provide a detailed overview of production and purification methods

Which culture medium was employed for Sortase A production?

Which sonication protocol was employed (details)?

How was the IMAC performed (details, buffers, linear gradient vs step gradient)?

Which column was employed for SEC and in which buffer?

Comment 2.5 (Pages 2 and 3, Lines 87 to 91): please provide a detailed overview of production and purification methods

Which vector was employed?

Which transfection protocol was employed, what were the culturing conditions, when was the supernatant harvested for protein purification, how was the supernatant processed before purification (details)?

How was the Protein G purification performed (details, buffers, etc)?

Which column was employed for SEC and in which buffer?

Comment 2.6 (Page 3, Line 89): I think “Fc” is missing after IgG1?

Comment 2.7 (Page 3, Lines 96 and 99): The information here is redundant. Would it be possible to state this information once at the end of the paragraph? For example, “In both cases, unreacted nanobody…”

Comment 2.8 (Page 3, Line 116): this is the first time “RLU” is mentioned so please define what RLU stands for.

Comment 2.9 (Page 3, Line 116): “nanobodu” should be “nanobody”

COMMENTS RELATED TO RESULTS

Comment 3.1 (Page 4, Line 137): please move the word “site-specifically” in front of “install” so that it reads as follows “to site-specifically install”

Comment 3.2 (Page 4, Line 138): please include “the generation of multivalent Ty1” between “for” and “multivalent”.

Comment 3.3 (Page 4, Line 140): I think it will be nicer to combine the first two sentences into one sentence, for example “First, we functionalized nanobody Ty1 (Fig. 1A) by attaching a dibenzocyclooctyne-amine (DBCO-NH2) to the C-terminus of Ty1”.

Comment 3.4 (Page 4, Line 148): please include “the generation of” between “for” and “various”.

Comment 3.5 (Page 5, Figure 1): I think this figure is visually very strong and is excellent to ensure that the readers fully grasp what is going on. The only suggestion I have would be to display the His-tag instead of XX in panel A to integrate this figure even better into the story that is being presented in this specific paper (removal of the His-tag is also mentioned in the figure legend, but the figure does not show a His-tag).

Comment 3.6 (Page 5, Line 153): full stop missing after “constructs”.

Comment 3.7 (Page 5, Line 162): one full stop too many.

Comment 3.8 (Page 6, Figure 2): please show the molecular weight marker and mention which marker was used in this experiment. Now, the molecular weights of the marker are displayed, but showing the marker is an essential part of presenting a protein gel in my opinion.

Comment 3.9 (Page 6, Figure 2): I would like to see the profiles of the gel filtrations that were performed to obtain the various fragments presented in Figure 2. I think this is important as the authors state that the proteins don’t fully multimerize after the chemistry, but that multimers can be separated from each other and monomers via gel filtration. For other scientists in the field attempting to perform similar work, it would be nice to have some reference profiles (“this is my result, what did the authors in this paper observe?”). I would thus like to see the gel filtration profiles for:

Ty1

Ty1-Ty1

Ty1-PEG11-Ty1

Ty1-Fc

4-arm PEG-Ty1

Please include which columns were used, which buffers were employed and please include the SDS-PAGE analysis of the fractions under the relevant peaks to demonstrate that the distinct multimeric species were indeed separated from each other. This Figure can be a supplemental figure.

Comment 3.10 (Page 6, Line 190): I believe there should be a comma after “SARS-CoV-2”

Comment 3.11 (Page 6, Line 192): please remove the word “by” after IC50.

Comment 3.12 (Page 6, Line 199): please replace “IC50s” by “IC50 values”.

Comment 3.13 (Page 7, Line 203): “Figure 1” should be “Figure 3”

Comment 3.14 (Page 7, Figure 3): given that this paper is visually quite strong since the authors have generated nice schematics of the different Nb constructs employed, I was wondering whether it would be possible to provide a schematic of the experimental set-up. This is not mandatory, but could be worth considering.

Comment 3.15 (Page 7, Line 215): I would replace “extremely” by a softer statement such as “a significantly enhanced”

Comment 3.16 (Suppl Fig): The nomenclatures of the constructs and the colors are not consistent throughout the different panels and not consistent with Figure 3 in the main text. I think this should be adapted accordingly.

Comment 3.17, overall: Also, the nomenclature of all the constructs don’t seem to be consistent throughout the entire manuscript and all figures. For instance, the Ty1-PEG dimer is also referred to “Ty1-PEG-Ty1” and “Ty1-PEG11-Ty1”. Please have another look and ensure that all names are consistently the same throughout the manuscript such to minimize possible confusion for the readers.

Comment 3.18 (Suppl Fig): Why was Ty1-Fc not included in the experiments presented in panel B?

COMMENTS RELATED TO DISCUSSION

Comment 4.1 (Page 7, Line 226): “from” should be replaced by “of”.

Comment 4.2 (Page 7, Lines 228 to 229). I think it would be good to expand the discussion a bit starting from this statement and supply more references.

Comment 4.3 (Page 8, Line 246): please replace “faster” by “relatively fast”

Comment 4.4, general: it is not really mentioned why these multivalent constructs are generated. Is it for diagnostic purposes? Therapeutic purposes? If for the latter, I think it would be interesting to make a note on the immunogenicity of such constructs (especially the 4-arm PEG) and provide some references.

Thank you for bearing with me through all of these comments. As I mentioned, all of them are meant to be as constructive as possible. I would like to congratulate the authors once again for their work and wish them the best of luck with the revisions.

Author Response

ARTICLE SUMMARY

In this paper, Moliner-Morro and colleagues present the generation of multivalent constructs of Nanobody Ty1, a camelid single-domain antibody directed against the spike protein of SARS-CoV-2. In order to produce these multivalent constructs, the authors employ the sortase A tag-mediated ligation followed by click chemistry. The Ty1-derived constructs are then tested for their potency to interfere with viral infectivity and the authors successfully demonstrate the improved potency of the multivalent constructs.

DETAILED REPORT – OVERALL CONSIDERATION

I would like to congratulate the authors with the amount of work they have performed and with the care they have taken to compile this manuscript. Overall, the paper is well-written and the work appears to be technically well-executed. The manuscript presents novel, interesting findings that are supported by nice figures. However, some portions of the manuscript remain on the “light” side and more effort should be put into expanding certain sections. I think this will only further improve the quality of the manuscript and provide the interested readers with a very complete description of both the literature and the results. My recommendation would be to accept this manuscript after a round of minor revisions. My comments/suggestions are intended to be as constructive as possible and I hope the authors will read them bearing this in mind.

We thank the reviewer for the thorough and constructive review of our manuscript.

COMMENTS RELATED TO THE ABSTRACT AND INTRODUCTION

Comment 1.1 (Page 1, Line 29): I believe there should be a comma after “antibodies”.

Done

Comment 1.2 (Page 1, Line 31): The reference molecular mass of Nanobodies is usually ~15 kDa. Please adapt accordingly.

Done

Comment 1.3 (Page 1, Line 35): “E.coli” seems to be lacking a space between the “E.” and “coli” + the following comma and “or” should not be italicized.

These errors have been fixed

Comment 1.4 (Page 1, Line 39): I believe there should be a comma after “agents”.

Done

Comment 1.5 (Page 2, Lines 44 to 45): The authors rightfully state that multi-valency is often achieved through the generation of Nb-IgG Fc fusions, but there are other methods as well and I think the authors should expand on this to make the introduction more complete. Examples include:

Fusing Nbs to the B-subunit of E. coli verotoxin.

Zhang et al 2004 (doi: 10.1016/j.jmb.2003.09.034)

Generating multi-valent Nb-Nb fusions

Pinto Torres et al 2018 (doi: 10.1038/s41598-018-26732-7)

Bernedo-Navarro et al 2018 (doi: 10.3390/toxins10030108)

Desmyter et al 2017 (doi: 10.3389/fimmu.2017.00884)

Generating multi-valent Nb-Nb-Ig Fc fusions

Laursen et al 2018 (doi: 10.1126/science.aaq0620)

The authors should please feel free to include other relevant papers to their liking.

We thank the reviewer for this suggestion. We expanded the introduction accordingly and included the suggested references.

Comment 1.6 (Page 2, Lines 46 to 54): Also, here, I propose to expand the introduction on the use of sortase A. Many sortase A – Nb papers have been published in recent years and I feel that the authors could provide a bit more context.

Examples include

Hagemeyer et al 2015 (doi: 10.1038/nprot.2014.177)

Massa et al 2016 (doi: 10.1002/cmmi.1696)

Besides referencing to reviews that discuss sortase A in detail, we now also include a few more examples of nanobody labeling by sortase A in the introduction.

COMMENTS RELATED TO MATERIALS AND METHODS

With regards to the production and purification procedures, I feel the authors are too restrictive and should provide more experimental details. This will provide interested readers with the full information to reproduce these experiments and appreciate how much work went into this. There is a general trend in research papers where detailed production and purification protocols are too often omitted. Producing and purifying correctly folded and functionally relevant proteins lies at the basis of our craft and requires a lot of hands-on expertise and experience. The inclusion of detailed protocols will better reflect the amount and quality of work performed by the authors.

We agree with the reviewer that more detail is required and have corrected that according to the constructuve suggestions below.

Comment 2.1 (Page 2, Line 72): given that protein purification is also discussed here I would propose to rename the title to “Protein production and purification”.

Done

Comment 2.2 (Page 2, Lines 73-80): please provide a detailed overview of production and purification methods

Which culture medium was employed for Ty1 production?

How was the periplasmic extraction performed (details)?

How was the IMAC performed (details, buffers, linear gradient vs step gradient)?

Which column was employed for SEC?

We now provide more detailed information on expression and purification.

Comment 2.3 (Page 2, Line 76): “E.coli” seems to be lacking a space between the “E.” and “coli”

Done 

Comment 2.4 (Page 2, Lines 81 to 86): please provide a detailed overview of production and purification methods

Which culture medium was employed for Sortase A production?

Which sonication protocol was employed (details)?

How was the IMAC performed (details, buffers, linear gradient vs step gradient)?

Which column was employed for SEC and in which buffer?

We now provide more detailed information on expression and purification.

Comment 2.5 (Pages 2 and 3, Lines 87 to 91): please provide a detailed overview of production and purification methods

Which vector was employed?

Which transfection protocol was employed, what were the culturing conditions, when was the supernatant harvested for protein purification, how was the supernatant processed before purification (details)?

How was the Protein G purification performed (details, buffers, etc)?

Which column was employed for SEC and in which buffer?

We now provide more detailed information on expression and purification.

Comment 2.6 (Page 3, Line 89): I think “Fc” is missing after IgG1?

Done

Comment 2.7 (Page 3, Lines 96 and 99): The information here is redundant. Would it be possible to state this information once at the end of the paragraph? For example, “In both cases, unreacted nanobody…”

We like this suggestion and have adjusted the text accordingly.

Comment 2.8 (Page 3, Line 116): this is the first time “RLU” is mentioned so please define what RLU stands for.

Done

Comment 2.9 (Page 3, Line 116): “nanobodu” should be “nanobody”

Done

COMMENTS RELATED TO RESULTS

Comment 3.1 (Page 4, Line 137): please move the word “site-specifically” in front of “install” so that it reads as follows “to site-specifically install”

Much better! Thank you!

Comment 3.2 (Page 4, Line 138): please include “the generation of multivalent Ty1” between “for” and “multivalent”.

Done

Comment 3.3 (Page 4, Line 140): I think it will be nicer to combine the first two sentences into one sentence, for example “First, we functionalized nanobody Ty1 (Fig. 1A) by attaching a dibenzocyclooctyne-amine (DBCO-NH2) to the C-terminus of Ty1”.

Done

Comment 3.4 (Page 4, Line 148): please include “the generation of” between “for” and “various”.

Done

Comment 3.5 (Page 5, Figure 1): I think this figure is visually very strong and is excellent to ensure that the readers fully grasp what is going on. The only suggestion I have would be to display the His-tag instead of XX in panel A to integrate this figure even better into the story that is being presented in this specific paper (removal of the His-tag is also mentioned in the figure legend, but the figure does not show a His-tag).

We have adjusted the figure accordingly.

Comment 3.6 (Page 5, Line 153): full stop missing after “constructs”.

Done

Comment 3.7 (Page 5, Line 162): one full stop too many.

Done

Comment 3.8 (Page 6, Figure 2): please show the molecular weight marker and mention which marker was used in this experiment. Now, the molecular weights of the marker are displayed, but showing the marker is an essential part of presenting a protein gel in my opinion.

We now show the marker in Figure 2.

Comment 3.9 (Page 6, Figure 2): I would like to see the profiles of the gel filtrations that were performed to obtain the various fragments presented in Figure 2. I think this is important as the authors state that the proteins don’t fully multimerize after the chemistry, but that multimers can be separated from each other and monomers via gel filtration. For other scientists in the field attempting to perform similar work, it would be nice to have some reference profiles (“this is my result, what did the authors in this paper observe?”). I would thus like to see the gel filtration profiles for:

Ty1

Ty1-Ty1

Ty1-PEG11-Ty1

Ty1-Fc

4-arm PEG-Ty1 

Please include which columns were used, which buffers were employed and please include the SDS-PAGE analysis of the fractions under the relevant peaks to demonstrate that the distinct multimeric species were indeed separated from each other. This Figure can be a supplemental figure.

We now show the elution profiles of the multimers that were run on the same size exclusion column: Ty1-dimer, Ty1 PEG dimer, Ty1 Fc and the 4 arm PEG in supplementary figure 1. We indicate the collected fractions that were used for the gel in Fig 2 as well as all neutralization assays.

Comment 3.10 (Page 6, Line 190): I believe there should be a comma after “SARS-CoV-2”

Done

Comment 3.11 (Page 6, Line 192): please remove the word “by” after IC50.

Done

Comment 3.12 (Page 6, Line 199): please replace “IC50s” by “IC50 values”.

Done

Comment 3.13 (Page 7, Line 203): “Figure 1” should be “Figure 3”

Already fixed by the editors.

Comment 3.14 (Page 7, Figure 3): given that this paper is visually quite strong since the authors have generated nice schematics of the different Nb constructs employed, I was wondering whether it would be possible to provide a schematic of the experimental set-up. This is not mandatory, but could be worth considering.

We thank the reviewer for this suggestion, but we do not believe that an additional schematic would significantly improve the manuscript.

Comment 3.15 (Page 7, Line 215): I would replace “extremely” by a softer statement such as “a significantly enhanced”

We have adjusted the text based on this suggestion.

Comment 3.16 (Suppl Fig): The nomenclatures of the constructs and the colors are not consistent throughout the different panels and not consistent with Figure 3 in the main text. I think this should be adapted accordingly.

We now have adjusted the figures accordingly.

Comment 3.17, overall: Also, the nomenclature of all the constructs don’t seem to be consistent throughout the entire manuscript and all figures. For instance, the Ty1-PEG dimer is also referred to “Ty1-PEG-Ty1” and “Ty1-PEG11-Ty1”. Please have another look and ensure that all names are consistently the same throughout the manuscript such to minimize possible confusion for the readers.

We adjusted figures and text accordingly.

Comment 3.18 (Suppl Fig): Why was Ty1-Fc not included in the experiments presented in panel B?

There is no specific reason. There was simply not sufficient space on the culture plates to also include Ty1-Fc and test its effect on already infected cells.

COMMENTS RELATED TO DISCUSSION

Comment 4.1 (Page 7, Line 226): “from” should be replaced by “of”.

Done

Comment 4.2 (Page 7, Lines 228 to 229). I think it would be good to expand the discussion a bit starting from this statement and supply more references.

We have expanded the discussion. We currently do not have data on the detailed mechanism that explain the extreme potency. While we have experiments in mind that elucidate this question using cryo EM tomography, those experiments will take considerable time and effort to perform. We would prefer to not speculate on the exact mechanism without further data. 

Comment 4.3 (Page 8, Line 246): please replace “faster” by “relatively fast”

Done

Comment 4.4, general: it is not really mentioned why these multivalent constructs are generated. Is it for diagnostic purposes? Therapeutic purposes? If for the latter, I think it would be interesting to make a note on the immunogenicity of such constructs (especially the 4-arm PEG) and provide some references.

This manuscript describes the proof-of-principle generation of multivalent nanobody constructs and we show that they work beautifully for in vitro neutralization of a virus. As mentioned in the discussion, the suitability (e.g. serum half-life etc) and potential application as a therapeutic still needs to be validated. We prefer not to speculate too much without having generated further data.  

Thank you for bearing with me through all of these comments. As I mentioned, all of them are meant to be as constructive as possible. I would like to congratulate the authors once again for their work and wish them the best of luck with the revisions.

Thank you!

Reviewer 2 Report

In the present manuscript by Moliner-Morro, Sheward et al. present different oligomeric variants of Ty1 nanobody that binds SARS2 spike protein. Dimers and tetramers were prepared by covalently linking C-termini of nanobodies via sortase A reaction and strain promoted azide-alkyne click chemistry. These constructs exhibit increased affinity in the picomolar range and are potentially useful for virus neutralization. The manuscript is well written, the results are presented clearly, therefore I recommend this article for publication with some minor revisions. Several minor issues are listed below:

  1. Introduction

Second paragraph of the introduction (Lines 39-45) would benefit from some references (differences between nanobodies and conventional antibodies, nanobody fusions etc.)

  1. Methods

Several points could be explained in more detail. Eg:

Line 101- What were the concentrations of proteins in the reaction? Was buffer same one as for functionalization of Ty1 with DBCO?

Line 104- What were the reaction conditions that yielded PEG-Ty1 dimers and tetramers (temperature, time, protein concentrations used)? For bis-dPEG11-DBCO we learn that it was added in several steps -how many? What about tetramer with 4-arm PEG-azide? - there is no information how this construct was prepared

  1. Results, figures

-Figures are incorrectly numbered- ie. Figure 3 is numbered as Fig. 1

-Figure 3 (neutralization of virus). I believe that in panel B points for the black curve (Ty1-Fc) are missing. (also in the SI!)

Author Response

In the present manuscript by Moliner-Morro, Sheward et al. present different oligomeric variants of Ty1 nanobody that binds SARS2 spike protein. Dimers and tetramers were prepared by covalently linking C-termini of nanobodies via sortase A reaction and strain promoted azide-alkyne click chemistry. These constructs exhibit increased affinity in the picomolar range and are potentially useful for virus neutralization. The manuscript is well written, the results are presented clearly, therefore I recommend this article for publication with some minor revisions. Several minor issues are listed below:

We thank the reviewer for the positive evaluation of our manuscript and the constructive suggestions below.

  1. Introduction

Second paragraph of the introduction (Lines 39-45) would benefit from some references (differences between nanobodies and conventional antibodies, nanobody fusions etc.)

We have now expanded the introduction and added relevant references.

     2. Methods

Several points could be explained in more detail. Eg:

Line 101- What were the concentrations of proteins in the reaction? Was buffer same one as for functionalization of Ty1 with DBCO?

We indicate the protein concentrations in µM, and now specify the reaction buffers.

Line 104- What were the reaction conditions that yielded PEG-Ty1 dimers and tetramers (temperature, time, protein concentrations used)? For bis-dPEG11-DBCO we learn that it was added in several steps -how many? What about tetramer with 4-arm PEG-azide? - there is no information how this construct was prepared

We now added the experimental details for these reactions.

     3. Results, figures

-Figures are incorrectly numbered- ie. Figure 3 is numbered as Fig. 1

The editorial office has already corrected this error.

-Figure 3 (neutralization of virus). I believe that in panel B points for the black curve (Ty1-Fc) are missing. (also in the SI!)

We have corrected this error in Figure 3 and Figure S2.

Reviewer 3 Report

Review for Picomolar SARS-CoV-2 neutralization using multi-arm PEG nanobody constructs

The authors describe their work to synthesize and assess multivalent versions of a nanobody that binds to the SARS-CoV2 receptor binding domain (Ty2). Reports on techniques to improve the properties SARS-CoV2 targeting moieties are urgently important. The manuscript provides a clear and concise rationale (lower dissociation rates, higher affinity) for investigating the impact of multivalency on inhibitory properties. The method used for creating multivalent constructs primarily relies on site specific labeling at the nanobody C-terminus with Sortase A. This technology is well established and appropriately cited. The biochemical methods used for dimer and tetramer synthesis and purification are well described. The improved properties of the multivalent nanobody constructs relative to the monomeric nanobody are clearly shown from the viral neutralization assays presented. In total the data and methods are clear and compelling and well-suited for publication here.

Here I list a few points of clarification that would provide improved mechanistic insights and context:

*Constructs consisting of multiple copies of nanobodies (consisting of either one nanobody or multiple nanobodies) have been expressed and have shown useful biological activities (DOI: 10.1128/CVI.00730-15). Could you provide more of an discussion of the benefits and drawbacks of enzyme mediated dimerization vs expression of biological fusions?

*Do you propose that for dimers or tetramers that more than one RBD in the same trimeric assembly can be engaged at the same time by the nanobodies in the same construct? How does this track with the length of the PEG linkers and the known structural details of the RBD?

*Are the error bars for the Ty1 dimers too small to see in Figure 1? This should be noted.

*On page 3 line 116 there is a typo: "nanobodu" instead of "nanobody"

Author Response

The authors describe their work to synthesize and assess multivalent versions of a nanobody that binds to the SARS-CoV2 receptor binding domain (Ty2). Reports on techniques to improve the properties SARS-CoV2 targeting moieties are urgently important. The manuscript provides a clear and concise rationale (lower dissociation rates, higher affinity) for investigating the impact of multivalency on inhibitory properties. The method used for creating multivalent constructs primarily relies on site specific labeling at the nanobody C-terminus with Sortase A. This technology is well established and appropriately cited. The biochemical methods used for dimer and tetramer synthesis and purification are well described. The improved properties of the multivalent nanobody constructs relative to the monomeric nanobody are clearly shown from the viral neutralization assays presented. In total the data and methods are clear and compelling and well-suited for publication here.

We thank the reviewer for the positive evaluation of our manuscript and the constructive suggestions below.

Here I list a few points of clarification that would provide improved mechanistic insights and context:

*Constructs consisting of multiple copies of nanobodies (consisting of either one nanobody or multiple nanobodies) have been expressed and have shown useful biological activities (DOI: 10.1128/CVI.00730-15). Could you provide more of an discussion of the benefits and drawbacks of enzyme mediated dimerization vs expression of biological fusions?

We now expanded the discussion according to this suggestion.

*Do you propose that for dimers or tetramers that more than one RBD in the same trimeric assembly can be engaged at the same time by the nanobodies in the same construct? How does this track with the length of the PEG linkers and the known structural details of the RBD?

Yes. Based on our cryo EM structure (presented in Hanke et al Nat Comms 2020), three Ty1 molecules can bind to one spike concurrently. We now clarified this in the discussion.

*Are the error bars for the Ty1 dimers too small to see in Figure 1? This should be noted.

We thank the reviewer for this comment. Some error bars were indeed missing, others too small to be visible. We corrected the error and included a note in the figure legend.

*On page 3 line 116 there is a typo: "nanobodu" instead of "nanobody"

Fixed